# The Importance of the Novel Postpartum Uterine Ultrasonographic Scale in Numerical Assessments of Uterine Involution Regarding Perinatal Maternal and Fetal Outcomes

**DOI:** 10.3390/diagnostics11091731

**Published:** 2021-09-21

**Authors:** Roxana Covali, Demetra Socolov, Alexandru Carauleanu, Ioana Pavaleanu, Mona Akad, Lucian Vasile Boiculese, Razvan Vladimir Socolov

**Affiliations:** 1Department of Radiology, Elena Doamna Obsterics and Gynecology University Hospital, 700398 Iasi, Romania; 2Department of Obstetrics and Gynecology, Cuza Voda Obstetrics and Gynecology University Hospital, 700038 Iasi, Romania; socolov@hotmail.com (D.S.); acarauleanu@yahoo.com (A.C.); 3Department of Obstetrics and Gynecology, Elena Doamna Obstetrics and Gynecology University Hospital, 700398 Iasi, Romania; ioana-m-pavaleanu@umfiasi.ro (I.P.); socolovr@yahoo.com (R.V.S.); 4Department of Obstetrics and Gynecology, Grigore T. Popa University of Medicine and Pharmacy, 700115 Iasi, Romania; akad.mona@yahoo.com; 5Department of Statistics, Grigore T. Popa University of Medicine and Pharmacy, 700115 Iasi, Romania; lboiculese@gmail.com

**Keywords:** uterine ultrasonography, obstetric delivery, postpartum hemorrhage, uterine retraction, endometrial length

## Abstract

Background: Uterine involution assessments are critical for the prevention of postpartum hemorrhage. Various methods have been used worldwide. Methods: The PUUS (Postpartum Uterine Ultrasonographic Scale) method evaluates, by transabdominal ultrasonography, the length of the endometrium of the uterine cavity occupied by blood or debris, from grade 0 (no blood) to grade 4 (over three-quarters of the endometrial length occupied by blood/debris). A total of 131 consecutive patients admitted for delivery in the Elena Doamna Obstetrics and Gynecology University Hospital in Iasi, Romania, were prospectively evaluated using the PUUS method. The mean age was 27.72 years old, and they were examined during the first 24–48 h after vaginal delivery, or in the first 48–72 h after cesarean delivery. For patients with a PUUS grade greater than 1, re-examination was preformed daily in the following days, until the PUUS grade decreased to 1 or 0. Results: By standardizing uterine involution in a numerical fashion, we precisely demonstrate that uterine involution varied with the method of delivery (vaginal/cesarean) and with the number of vials of oxytocin received intrapartum, but not with the number of vials of ergometrine maleate received, and not with the origin of the parturient (rural/urban).

## 1. Introduction

The most common and severe complications in the post-placental phase are bleeding and disturbances of uterine involution (Schrey) [1]. Various methods can be used to evaluate the uterine postpartum cavity: abdominal and/or transvaginal ultrasonography, focused assessment with sonography for obstetrics, two- and three-dimensional ultrasound examinations, contrast-enhanced ultrasound, computed tomography (CT) scans [2], magnetic resonance imaging (MRI) [3], or even transabdominal and transvaginal sonography with magnetic resonance imaging fusion. Measurements can be made using these various approaches for assessing the uterine cavity diameters, uterine volume, the lower uterine segment thickness, the posterior or anterior uterine wall, cesarean scarring, gas in the uterine cavity, etc. However, these approaches cannot be used routinely. In addition, there is no consensus on a standardized method for postpartum ultrasound; therefore, more research and standardization are necessary, according to De Winter. [4] 

Kostrubiak [5] emphasized that ultrasonography is the mainstay in the initial imaging evaluation of postpartum patients, with the occasional progression to CT, MRI, or angiography. Therefore, we focused on ultrasonographic examinations of the postpartum patients.

In a systematic review, in women who underwent transabdominal or transvaginal ultrasound from the first postpartum day to a maximum of 6 weeks postpartum, Ucci [6] found that the upper limit of normal (95% percentile) for the endometrial thickness was 25 mm by 7 days postpartum, and this continued to decrease in a similar manner for all the women, regardless of parity or the mode of delivery. According to Ucci, these upper limits of the normal postpartum endometrium and uterine measurements in uncomplicated pregnancies provide clinical guidance for the sonographic evaluation of women with complicated postpartum courses. On the other hand, Vyas [7] demonstrated that an endometrial thickness under 10 mm, an avascular endometrium, and an absence of enhanced myometrial vascularity are the sonographic features associated with successful noninvasive management for postpartum hemorrhage or suspected retained products of conception. There is a considerable difference between a 25 mm and 10 mm endometrial thickness. Moreover, endometrial measurements are not sufficient in the postpartum evaluation of uterine cavities because endometrial measurements are performed toward the fundus of the uterus, and the opposite part may have blood or debris. Gill [8] showed that the fundal region may be well contracted although the lower uterine segment is dilated and atonic; therefore, bedside obstetric ultrasound imaging to reveal an echogenic endometrial stripe is an essential examination. The endometrial stripe ultrasonographic examination was standardized in our study.

The Committee on Practice Bulletins-Obstetrics [9] encourages obstetric care providers to play key roles in implementing standardized bundles of care (policies, guidelines, and algorithms) for the management of postpartum hemorrhage. Ibanez [10] evaluated the implementation of a maternal early warning system for monitoring patients during the first two hours after delivery; the parameters collected included uterine involution and bleeding, and demonstrated low sensitivity, high specificity, with a significant number of false negatives. Thus, more research was required. Cole [11] evaluated, by palpation, and not by imaging, the 0 to 10 numeric rating scale for uterine tone, and showed that it could be a reliable, standardized tool in reporting the degree of uterotonic contraction during cesarean delivery. Although it was a standardized scale, it could be used only in cesarean deliveries, not in vaginal deliveries, and only during labor, not after—not to mention that no imaging method was involved. The postpartum uterine ultrasonographic scale emerged as fulfilling the need for a standardized tool of assessing uterine postpartum involution, which can be used in all patients, no matter the method of delivery, and in all ranges of medical establishments, with only basic equipment (abdominal ultrasound) available.

Cesarean rates have increased worldwide [12]. Cilingir [13] determined sonographic findings of the postpartum uterus 24 h after vaginal and cesarean deliveries and discovered that both fundus–cervix length and promontorium–fundus length were significantly greater in the vaginal delivery group compared with the cesarean delivery group. However, no research has been conducted after the first 24 h between vaginal delivery and cesarean delivery groups.

Several uterotonic drugs are available for preventing postpartum hemorrhage; drugs that reduce blood loss are routinely recommended. Salati [14] concluded that prophylactic oxytocin for the third stage of labor, compared with no uterotonics, may reduce blood loss, but the effect of oxytocin compared with ergot alkaloids is uncertain with regard to blood loss, the need for additional uterotonics, and blood transfusion. No study has compared the uterine dimensions after oxytocin or ergometrine administration.

Amjad [15] reported that compared with urban/high socioeconomic status mothers, rural/low socioeconomic status mothers had increased odds of postpartum hemorrhage and adverse pregnancy outcomes. However, no measurements have been made to evaluate the exact differences in uterine involution between the two groups, regarding maternal residence. 

The goal of this study was to use the novel PUUS scale in order to assess the uterine cavity involution depending on maternal and fetal outcomes, on the method of delivery and medication received.

## 2. Materials and Methods

The PUUS method is a visual scale to evaluate the number of quarters of the endometrial length occupied by blood or debris, ranging from 0 to 4. The PUUS name is an abbreviation from the novel Postpartum Uterine Ultrasonographic Scale, and it uses transabdominal ultrasonography. After standardizing the uterine involution in a numerical fashion, the objectives of this study are as follows: (1) to analyze the uterine involution of patients who underwent cesarean delivery compared with patients who delivered vaginally, not only during the first postpartum day, but also during the following days; (2) to analyze the uterine cavity after oxytocin or ergometrine maleate administration; (3) to study uterine involution, regarding maternal residence (urban/rural); and (4) to compare uterine involution after postpartum uterine instrumental control with that after manual control or no control. 

The PUUS method evaluates the proportion of the endometrial length occupied by blood or debris, as follows (Figure 1, Figure 2, Figure 3, Figure 4 and Figure 5): 

Grade 0: no blood or debris in the uterine cavity;

Grade 1: less than a one-quarter of the endometrial length occupied by blood or debris;

Grade 2: less than one-half of the endometrial length occupied by blood or debris;

Grade 3: less than three-quarters of the endometrial length occupied by blood or debris;

Grade 4: over three-quarters of the endometrial length occupied by blood or debris.

This study was approved by the Ethics Committee of Elena Doamna Obstetrics and Gynecology University Hospital (approval number: 9, 15 September 2017). Informed written consent was obtained from each patient.

In the entire prospective study, 160 consecutive Caucasian patients, who delivered in an obstetrics and gynecology university hospital, from October 2017 until December 2017, either by cesarean delivery or by vaginal delivery, were considered for evaluation using the PUUS method. After excluding 29 patients (as described below), 131 patients were included in our study.

Inclusion criteria for the study were as follows: (1) patients who delivered at 34 weeks of pregnancy or more, either by cesarean delivery or by vaginal delivery; (2) patients who delivered in our hospital; and (3) patients who had no infectious/highly contagious health problems.

Exclusion criteria were as follows: (1) patients who delivered under 34 weeks of pregnancy (hospital protocol requires them to be sent to a higher rank obstetrics and gynecology hospital for very fragile neonate health supportive reasons); and (2) patients who delivered at home and were admitted to our hospital to undergo a health check for themselves and their newborns (18 patients)—after being admitted, these patients were sent to a septic department of the hospital and were examined by other professionals, not by the authors of this study; and (3) patients who had infectious/highly contagious health problems (11 patients)—after delivery, these patients were sent to a septic department of the hospital and were examined by other professionals, not by the authors of this study.

Hospital policies require active management during the third stage of labor. Active management of the third stage of labor involves giving a prophylactic uterotonic, early cord clamping, and controlled cord traction to deliver the placenta. If required, patients received intravenous oxytocin and, sometimes, ergometrine maleate, during this third stage of labor. After delivery, patients were examined over the same time intervals: within the first 24–48 h after delivery in cases of vaginal delivery, and within the first 48–72 h in cases of cesarean delivery. To eliminate any potential sources of bias: (1) patients were examined in a random order, and the practitioner was unaware of the patient’s medical history at the time of the examination; and (2) there was only one practitioner (R.C.) who performed all the ultrasonography measurements, under the supervision of R.S.—evaluation was performed independently by both of them, the level of concordance was 95.41%, and the disagreements were solved by discussion.

As in the routine protocol, patients with PUUS grades 2, 3, or 4 were re-examined the following day. Mobility encouragement was used, and in cases when 48 h after delivery, on ultrasonographic abdominal examination, uterine cavity still contained blood or debris, with the image unchanged in the next three days, intramuscular oxytocin was injected on days 4, 5 and 6, twice a day, and ultrasonographic examination was performed daily until the PUUS score declined to 1 or 0.

Statistical analysis was performed using IBM SPSS v.18 (PASW Statistics for Windows, SPSS Inc., Chicago, IL, USA). For descriptive measures, we computed the mean, standard deviation, and minimum and maximum limits. Multiple group comparisons were performed using the ANOVA technique, whereas for two sets, the Student’s *t* test was applied. The ANOVA condition of variance homogeneity was assessed by using Levene’s test, and the Tamhane post hoc test was applied if ANOVA was not accepted; *p* = 0.05 was considered statistically significant. Only weight was normally distributed; for the other comparisons, nonparametric tests were used.

## 3. Results

### 3.1. Age of the Parturients

The mean age was 27.72 years (range, 15–42). The distribution of patients regarding the PUUS values is presented in Table 1.

The PUUS grade varied with age, but there was no statistically significant correlation between the PUUS grade and age of the parturients (*p* = 0.51) (Figure 6).

There was only one patient with a PUUS score of 4, and no statistical analysis could be made with this group; therefore, they were removed from the statistical analysis, and remain only in some tables of figures.

### 3.2. Method of Delivery

Forty-four (33.6%) patients delivered vaginally, and 87 (66.4%) had cesarean deliveries (Table 2).

The PUUS grade varied significantly (*p* = .002) with the method of delivery of the baby: vaginal delivery (0.84 ± 1.11, 95% CI: 0.5–1.18) or cesarean delivery (0.33 ± 0.71, 95% CI: 0.18–0.48). Vaginal delivery was associated with a significantly slower involution of the uterine cavity compared with cesarean delivery (Table 3).

The numbers and proportions of patients who delivered either by cesarean section or vaginally, for every value of PUUS, are detailed in Table 4. Most patients (69; 79.31%) delivered by cesarean section and had a PUUS grade of 0; the smallest group of patients (1; 2.27%) delivered vaginally and had a PUUS grade of 4.

Each one of the 87 patients who were offered cesarean section had at least one of the conditions presented in Table 5.

Regarding the macrosomia fetuses (*n* = 7; 5.34%), most of them (*n* = 5) were delivered by cesarean section, and some (*n* = 2) by vaginal delivery (Table 6). Macrosomia was considered when the fetal weight surpassed 4000 g, and it was not associated with high values of PUUS grade.

### 3.3. Origin of the Parturient

People with urban residency are generally considered as having higher socioeconomic status compared with people from rural areas, generally considered as having a lower socioeconomic status; therefore, we studied the patients’ residency as a reflection of the socioeconomic status. The PUUS grade varied according to the origin of the parturient: rural (0.56 ± 0.95; 95% CI: 0.37–0.77)) or urban (0.37 ± 0.75; 95% CI: 0.14–0.60) (Table 7).

In patients of rural origin (*n* = 88, 67.17%), uterine cavity involution was not significantly slower (*p* = .24) than in patients of urban origin (*n* = 43, 32.83%) (Table 8 and Table 9).

### 3.4. Oxytocin Vials

Patients received oxytocin (5-unit vials) and ergometrine maleate (0.2 mg vial) during the third stage of labor, when required by the active management. Regarding oxytocin, 8 (6.10%) patients received five vials each, 11 (8.39%) received four vials each, 22 (16.79%) received three vials each, 57 (43.51%) received two vials each, 21 (16.03%) received one vial each, and 12 (9.16%) received 0 vials, depending on the degree of blood loss (Table 10).

The PUUS grade varied and was inversely proportional (*p* = −.18) to the number of oxytocin vials received (Table 11 and Table 12).

### 3.5. Ergometrine Maleate

Regarding ergometrine maleate, 100 (76.33%) patients received 0 vials, 30 (22.90%) received one vial, and 1 (0.76%) received two vials, depending on the lack of uterine tonus, the amount of blood loss, and the lack of blood hypertension during the third stage of labor (Table 13).

The PUUS grade varied with the number of ergometrine maleate vials received. No significant difference existed between the groups (*p* = .28) (Table 14 and Table 15).

The PUUS score was constant when the number of vials of ergometrine maleate received was 2; therefore, this value was omitted from the table.

### 3.6. Cavity Control

Once the baby and the placenta were delivered, depending on the aspect and integrity of the placenta and the degree of blood loss, every gynecologist decided to perform either a manual control of the postpartum uterine cavity or instrumental control of the postpartum uterine cavity; alternatively, no control was required. Instrumental control was performed in 99 (75.57%) patients, manual control was conducted in 7 (5.34%), and no control was needed in 25 (19.08%) patients (Table 16).

The PUUS grade varied slightly with the presence and type of postpartum uterine control performed. However, there was no significant difference (*p* = .09) between the three groups, and even if the manual control group, consisting only of seven persons, were to be removed from the comparison, no statistically significant difference (*p* = .058) would occur between the instrumental control group and the no-control group (Table 17 and Table 18).

### 3.7. Birth Weight

The PUUS grade slightly varied with the birth weight (Figure 7).

### 3.8. Neonate Gender

The PUUS grade varied slightly with the gender of the neonate (Table 19).

There was only one value of PUUS = 4, and very few values of PUUS = 3 (6); therefore, they had to be removed from the statistical analysis. Considering only values of PUUS from 0 to 2, there was no statistical correlation between the PUUS and gender of the neonate (*p* = .70).

### 3.9. Neonate Apgar Score

The Apgar score was constant when PUUS = 4; therefore, it has been omitted from the statistical analysis. For PUUS scores of 0–3, there was no significant correlation between the PUUS grade and Apgar score of the neonate (*p* = .14) (Figure 8).

### 3.10. High Blood Pressure of The Parturient

The PUUS grade did not vary significantly with the presence/absence of high blood pressure of the parturient (*p* = .56) (Figure 9).

### 3.11. Placenta Previa

After removing the only case of PUUS = 4 and the few cases of PUUS = 3 (6 cases), the PUUS grade did not vary significantly with the presence/absence of placenta previa (*p* = .51) (Table 20). There were no patients with placenta previa and PUUS grade = 3 or PUUS grade = 4.

### 3.12. Bleeding

There was no case of heavy bleeding (severe postpartum hemorrhage, over 1000 mL) in this group of parturients. Most patients had a low level of bleeding (<500 mL), and a few had moderate bleeding (postpartum hemorrhage, >500 mL) (Table 21). There was no significant correlation between the level of bleeding and the PUUS grade (*p* = .44).

### 3.13. Gestational Age

There was no correlation between the gestational age and PUUS score (*p* = .30) (Table 22).

## 4. Discussion

Transabdominal sonography is suitable for examination of the uterus during the first 14 days postpartum, when the uterine body and position, as well as the cavity, are easy to examine by ultrasound. [16]. Uterine involution is rapid during the first week and then slows down [17]. Sonographic assessments of the uterine postpartum involution thus far involve various precise determinations: the size of the uterus and the uterine cavity measured on the longitudinal sections only [17], the thickness and length of the uterine cavity [18], the anteroposterior diameter of the uterus and uterine cavity [16,19], the uterine dimensions (height, length, width) and intracavitary thickness [20], the uterine area (π × 1/2 length × 1/2 width) [21], the uterine volume, calculated by the formula: longitudinal diameter (LD) × anteroposterior diameter (APD) × transverse diameter (TD) × 0.45 [18], or the distance between the uterine fundus–promontorium and uterine fundus–L5 [13]. The result of these methods is a mean value, whereas the result of the PUUS method was reported in quarters of the endometrial stripe of every single patient, which means it was flexible regarding the wide variability of patient dimensions. 

Frequent postpartum ultrasonographic findings include a thickened endometrial stripe and echogenic material in the uterine cavity [22], located in the cervical area in the early puerperum period [16]. Although the echogenic material commonly seen in the endometrial cavity of asymptomatic patients is not associated with the development of bleeding complications [16,22,23], it is the proportion of the endometrial stripe free of echogenic material that was assessed during this study, and the influence of the method of delivery, oxytocin administration, revision of the uterine cavity and socioeconomic status on this proportion.

Bardin et al. [20] and Fuller [24] found no sonographic differences in uterine involution according to the mode of delivery 24 h and 48 h from delivery, respectively. Based on the PUUS method, there was a difference regarding the method of delivery. In this study, 27 women were examined during the first 24 h postpartum. In the women who delivered vaginally (*n* = 23, PUUS grade 1.3), the uterine cavity area decreased more slowly than the uterine cavity in women who had a cesarean delivery (*n* = 4, PUUS grade 0). In our study, another 50 women were examined for the first time postpartum during the second day (24–48 h after delivery), and the same pattern remained: vaginal delivery (*n* = 10, PUUS grade 0.5) vs. cesarean delivery (*n* = 40, PUUS grade 0.15). As reported above, PUUS evaluation showed that there was a difference between vaginal delivery and cesarean delivery because vaginal delivery was accompanied by a slower involution of the uterine cavity. These findings were in accordance with Cilingir [13], who determined the sonographic findings of the postpartum uterus 24 h after delivery and demonstrated that the fundus–cervix length was significantly higher in the vaginal delivery group compared to the cesarean section group (*p* < 0.001). Our findings were also in accordance with Kristoschek [18] who, using ultrasonography, demonstrated that a significant difference occurred on day 2 postpartum, and showed that the uterus had a smaller volume following cesarean section compared with vaginal delivery (*p* = 0.04).

No prior study has analyzed uterine involution after postpartum uterine instrumental control compared to manual control or no control; this was assessed in the presented study. Bardin [20] found no clinically significant differences in uterine characteristics according to manual revision of the uterine cavity, although they did not also study the outcomes of the instrumental revision. We agree with Bardin: the PUUS grade varied slightly with the presence and type of postpartum uterine control performed, although there was no significant difference (*p* = .093) between the three groups (instrumental control, manual control, and no control groups) for all the patients who delivered both vaginally and by cesarean section. Berlit [25] reported that curettage in elective cesarean delivery was not beneficial. We report similar data: after cesarean delivery, instrumental control, not curettage, determined a PUUS grade of 0.4, higher than the manual control of the uterine cavity, which determined a PUUS grade of only 0.14, or no control (PUUS grade 0.2). This means that after cesarean deliveries, manual control generates uterine involution slightly faster than no control, and instrumental control is the least recommended. This would probably require a change in hospital protocol, to encourage manual control after cesarean delivery. Regarding vaginal delivery, instrumental control generated a PUUS grade of 0.75, whereas no control generated a PUUS grade of 0.95; these values were not significantly different. There was no manual control reported in vaginal deliveries during this study. This would probably require a change in hospital protocol as well, to discourage instrumental control in favor of no control after vaginal deliveries. We found no significant differences in uterine involution after instrumental control or manual control, as compared to no control.

Based on the information written on the identity cards and subsequently in the medical records, patients residing in villages were defined as coming from rural areas, whereas patients residing in towns or cities were categorized as originating from urban areas; people with an urban residency are generally considered to have a higher socioeconomic status than people from rural areas, who are generally considered as having a lower socioeconomic status. We found no significant differences in postpartum uterine involution regarding the patients’ residency, which reflected the socioeconomic status. 

In this related part of our study, we demonstrated that uterine involution varies with the method of delivery of the baby and with the number of vials of oxytocin administered intrapartum, but not with type of postpartum uterine control, the number of ergometrine maleate vials administered, or the origin of the parturient. 

This study considered only the dimensions of the uterine cavity, not the content, whether it was blood, debris, or nothing. Future studies are required to address this issue. Another major limitation is that we had very few patients with PUUS values of 3 or 4. Larger studies to examine the correlation between uterine involution and other specific factors (clinical characteristics of the patients, coagulopathies, uterine inversions, body mass index) with more patients are also required.

## 5. Conclusions

In this study, by standardizing the uterine involution in a numerical fashion (PUUS), we have precisely demonstrated that the uterine involution is slower in vaginal deliveries and increases with the number of vials of oxytocin received intrapartum. No significant influence was associated with any of the other conditions studied. This PUUS scale could further be used to study the influence of a variety of factors upon the speed of uterine involution and may improve the quality of postpartum care.

## Figures and Tables

**Figure 1 diagnostics-11-01731-f001:**
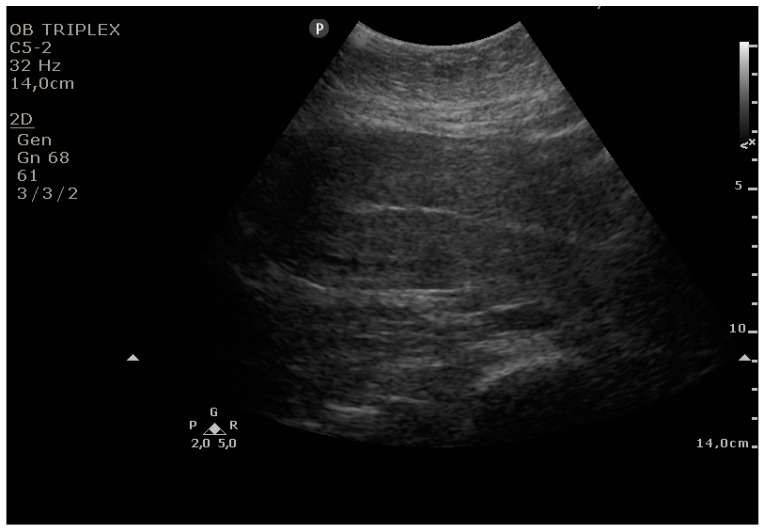
Grade 0: no blood or debris in the uterine cavity.

**Figure 2 diagnostics-11-01731-f002:**
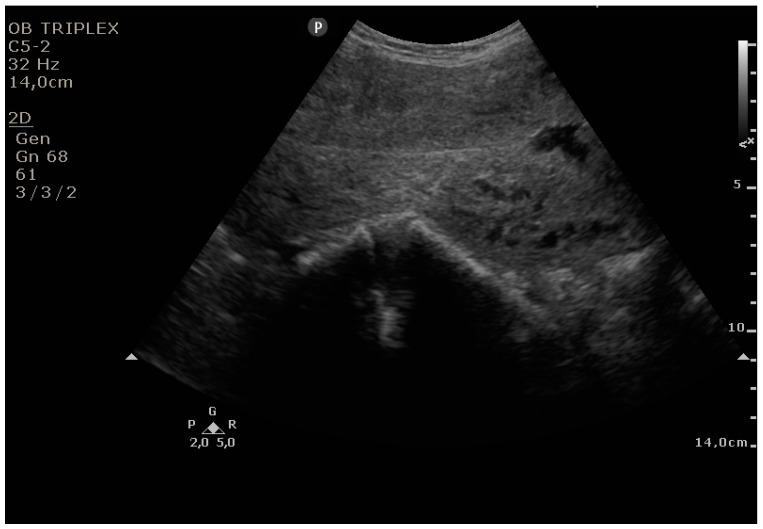
Grade 1: less than one-quarter of the endometrial length occupied by blood or debris.

**Figure 3 diagnostics-11-01731-f003:**
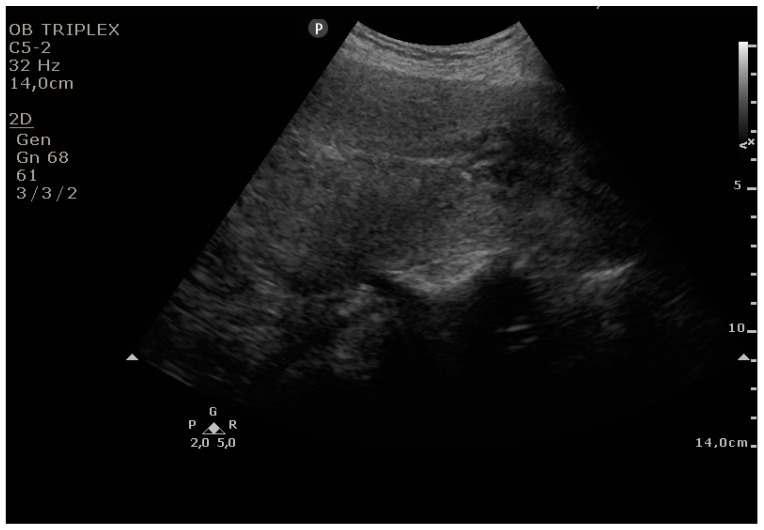
Grade 2: less than one-half of the endometrial length occupied by blood or debris.

**Figure 4 diagnostics-11-01731-f004:**
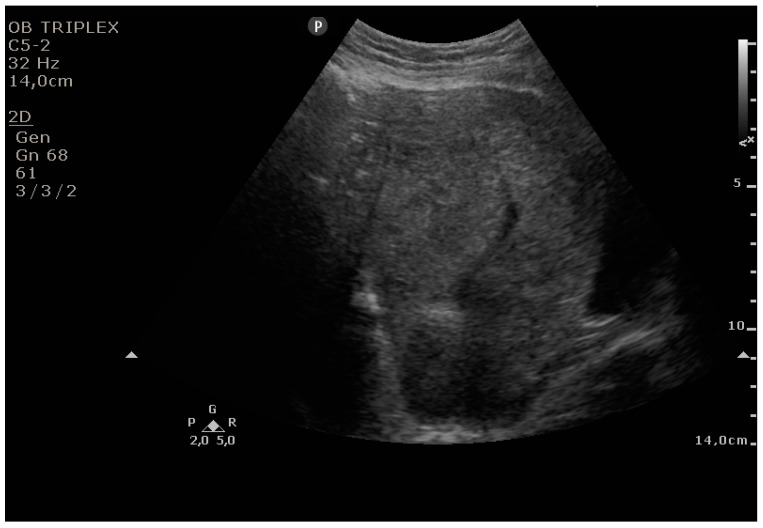
Grade 3: less than three-quarters of the endometrial length occupied by blood or debris.

**Figure 5 diagnostics-11-01731-f005:**
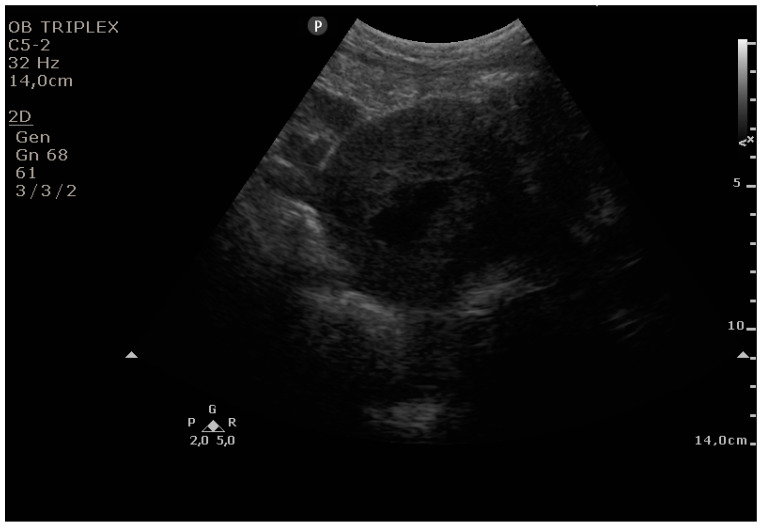
Grade 4: over three-quarters of the endometrial length occupied by blood or debris.

**Figure 6 diagnostics-11-01731-f006:**
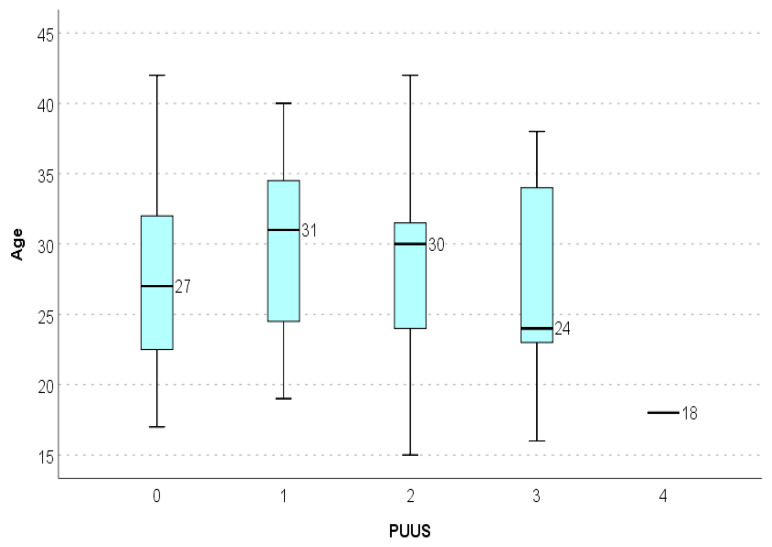
PUUS grade varied with age. (*p* = 0.51).

**Figure 7 diagnostics-11-01731-f007:**
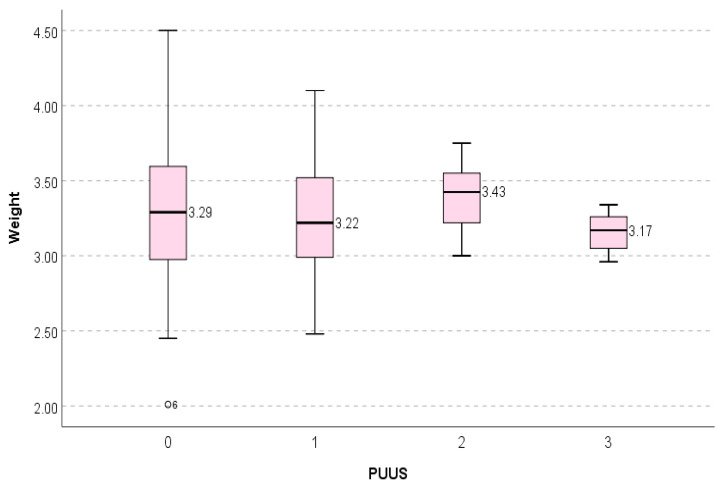
The PUUS grade did not vary significantly with the birth weight. Weight was constant when PUUS = 4; therefore, it was omitted from the statistical analysis.

**Figure 8 diagnostics-11-01731-f008:**
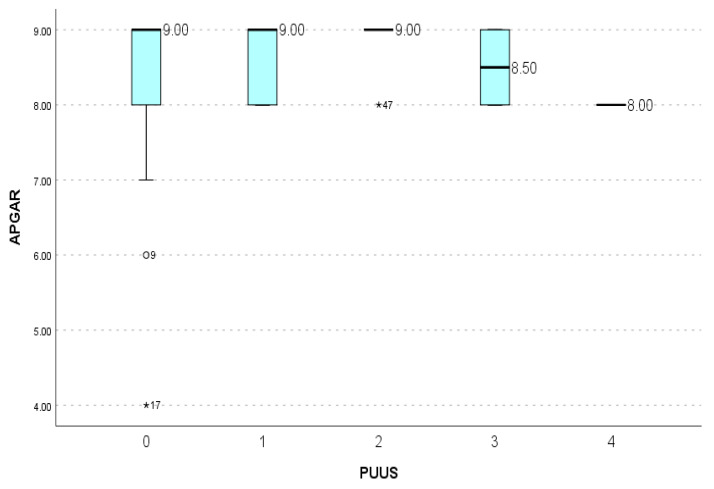
PUUS grades did not vary significantly with the Apgar scores of the neonate. * represents extreme outliers, data that markedly differ from other observations.

**Figure 9 diagnostics-11-01731-f009:**
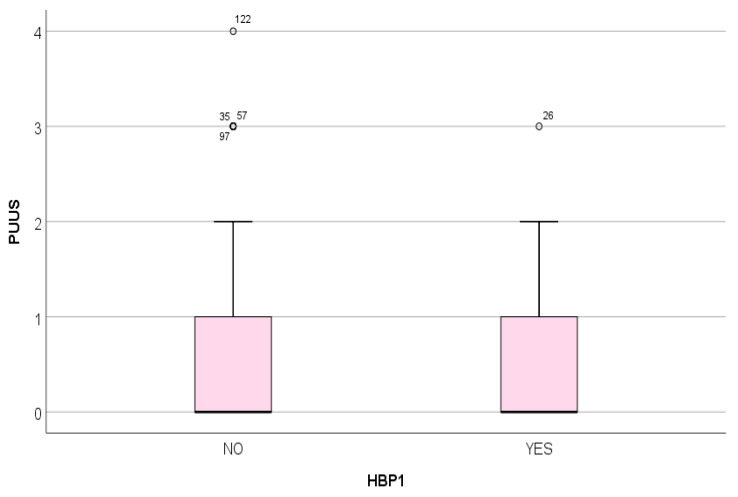
PUUS grade did not vary significantly with the presence (YES) or absence (NO) of high blood pressure of the parturient; HBP1, high blood pressure.

**Table 1 diagnostics-11-01731-t001:** Distribution of patients regarding the PUUS value.

PUUS Value	Number of Patients	Proportion
0	92	70.22%
1	20	15.26%
2	12	9.16%
3	6	4.58%
4	1	0.76%
Total	131	100%

**Table 2 diagnostics-11-01731-t002:** Distribution of patients regarding the method of delivery.

Method of Delivery	Number of Patients	Proportion
Cesarean	87	66.41%
Vaginal	44	33.58%

**Table 3 diagnostics-11-01731-t003:** The PUUS values of patients regarding the method of delivery (*p* = .002).

Method of Delivery	Mean Values	95% CI
Cesarean	0.33	0.18–0.48
Vaginal	0.84	0.50–1.18

**Table 4 diagnostics-11-01731-t004:** Proportion and number of patients for each PUUS value regarding the method of delivery.

PUUS	Cesarean (*n* = 87)	Vaginal (*n* = 44)
0	69 (79.31%)	23 (52.27%)
1	8 (9.2%)	12 (27.27%)
2	9 (10.34%)	3 (6.82%)
3	1 (1.15%)	5 (11.36%)
4	0 (0%)	1 (2.27%)

**Table 5 diagnostics-11-01731-t005:** Conditions that, single or associated, necessitated a cesarean section in the patients studied and the PUUS values for each condition.

Cesarean Indications	0	1	2	3	4	Total
Triggered at Term Birth	21	2	2	1	0	26
	80.76%	7.69%	7.69%	3.84%	0%	100%
Triggered Premature Birth	4	1	0	0	0	5
	80%	20%	0%	0%	0%	100%
Cesarean (Simple/Double) Scar Uterus	17	0	2	0	0	19
	89.47%	0%	10.52%	0%	0%	100%
Dystocia	8	3	2	1	0	14
	57.14%	21.42%	14.28%	7.14%	0%	100%
Imminence of Uterine Rupture	1	0	0	0	0	1
	100%	0%	0%	0%	0%	100%
Coiling of the Umbilical Cord (Simple/Double Coiling)	9	0	5	0	0	14
	64.28%	0%	35.71%	0%	0%	100%
Acute Fetal Intrauterine Distress	10	0	1	0	0	11
	90.90%	0%	9.09%	0%	0%	100%
Intrauterine Growth Restriction	2	0	0	0	0	2
	100%	0%	0%	0%	0%	100%
Placenta Previa	9	3	0	0	0	12
	75%	25%	0%	0%	0%	100%
Breech Presentation	5	0	0	0	0	5
	100%	0%	0%	0%	0%	100%
Brow Presentation	1	0	0	0	0	1
	100%	0%	0%	0%	0%	100%
Unsuccessful Trial of Labor	7	2	0	0	0	9
	77.77%	22.22%	0%	0%	0%	100%
Gestational Edema	5	0	1	0	0	6
	83.33%	0%	16.66%	0%	0%	100%
Painful Uterine Contractions	2	1	0	0	0	3
	66.66%	33.33%	0%	0%	0%	100%
Rh Incompatibility and Isoimmunization	1	1	0	0	0	2
	50%	50%	0%	0%	0%	100%
Preeclampsia	2	0	0	0	0	2
	100%	0%	0%	0%	0%	100%
Uterine Fibroids	2	0	0	0	0	2
	100%	0%	0%	0%	0%	100%
Oligoamnios	1	0	0	1	0	2
	50%	0%	0%	50%	0%	100%
Thrombophilia	1	0	0	0	0	1
	100%	0%	0%	0%	0%	100%
Splenectomized Parturient	1	0	0	0	0	1
	100%	0%	0%	0%	0%	100%
Lumbosciatica	1	0	0	0	0	1
	100%	0%	0%	0%	0%	100%
Hepatitis B	1	0	0	0	0	1
	100%	0%	0%	0%	0%	100%
Right UHN with JJ Stent and Reflux Pyelonephritis	1	0	0	0	0	1
	100%	0%	0%	0%	0%	100%
Left Buttock Abscess	1	0	0	0	0	1
	100%	0%	0%	0%	0%	100%
Total	113	13	13	3	0	142
	79.57%	9.15%	9.15%	2.11%	0%	100%

UHN, Ureterohydronephrosis.

**Table 6 diagnostics-11-01731-t006:** Distribution of patients with fetal macrosomia regarding the method of delivery and the PUUS value.

Macrosomia	PUUS 0	PUUS 1	PUUS 2	PUUS 3	PUUS 4	Total
Cesarean	4 (80.00%)	1 (20.00%)	0 (0.00%)	0 (0.00%)	0 (0.00%)	5 (100.00%)
Vaginal	1 (50.00%)	1 (50.00%)	0 (0.00%)	0 (0.00%)	0 (0.00%)	2 (100.00%)
Total	5 (71.42%)	2 (28.57%)	0 (0.00%)	0 (0.00%)	0 (0.00%)	7 (100.00%)

**Table 7 diagnostics-11-01731-t007:** Distribution of patients regarding their origin.

Origin of Patients	Number of Patients	Proportion
Rural	88	67.17%
Urban	43	32.82%

**Table 8 diagnostics-11-01731-t008:** The patients’ PUUS values regarding the origin of the parturients.

Method of Delivery	Mean Values	95% CI
Rural	0.57	0.37–0.77
Urban	0.37	0.14–0.60

**Table 9 diagnostics-11-01731-t009:** Number and proportion of patients for each PUUS value regarding the origin of the parturient.

PUUS	Rural (*n* = 88)	Urban (*n* = 43)
0	59 (67.04%)	33 (76.74%)
1	15 (17.04%)	5 (11.62%)
2	8 (9.09%)	4 (9.30%)
3	5 (5.68%)	1 (2.32%)
4	1 (1.13%)	0 (0%)

**Table 10 diagnostics-11-01731-t010:** Distribution of patients regarding the number of oxytocin vials received during the third stage of labor.

Number of Oxytocin Vials	Number of Patients	Proportion
0	12	9.16%
1	21	16.03%
2	57	43.51%
3	22	16.79%
4	11	8.39%
5	8	6.10%

**Table 11 diagnostics-11-01731-t011:** The PUUS values of patients regarding the number of oxytocin vials received during the third stage of labor.

PUUS Values	Mean Number of Oxytocin Vials Received	95% CI
0	2.33	2.06–2.60
1	1.80	1.20–2.40
2	2.00	1.46–2.54
3	1.67	1.12–2.21

The number of oxytocin vials was constant when PUUS = 4; therefore, it was omitted from this table.

**Table 12 diagnostics-11-01731-t012:** The PUUS values of patients regarding the number of oxytocin vials received during the third stage of labor.

PUUS Values	0	1	2	3	4
Number of Oxytocin Vials	-	-	-	-	-
1	66.67%	25.00%	8.33%	0.00%	0.00%
2	57.14%	23.81%	4.76%	9.52%	4.76%
3	66.67%	14.04%	12.28%	7.02%	0.00%
4	77.27%	9.09%	13.64%	0.00%	0.00%
5	90.91%	9.09%	0.00%	0.00%	0.00%

**Table 13 diagnostics-11-01731-t013:** Distribution of patients regarding the number of ergometrine maleate vials received during the third stage of labor.

Number of Ergometrine Maleate Vials	Number of Patients	Proportion
0	100	76.33%
1	30	22.90%
2	1	0.76%

**Table 14 diagnostics-11-01731-t014:** The PUUS values of patients regarding the number of ergometrine maleate vials received.

Number of Vials of Ergometrine Maleate	PUUS Values	95% CI
0	0.56	0.37–0.75
1	0.33	0.07–060

**Table 15 diagnostics-11-01731-t015:** The PUUS values of patients regarding the number of ergometrine maleate vials received.

Number of Ergometrine Maleate Vials	0	1	2	3	4
0	68.00%	15.00%	11.00%	5.00%	1.00%
1	76.66%	16.66%	3.33%	3.33%	0.00%
2	100%	0.00%	0.00%	0.00%	0.00%

**Table 16 diagnostics-11-01731-t016:** Distribution of patients regarding the type of uterine cavity control.

Type of Uterine Control	Number of Patients	Proportion
Instrumental	99	75.57%
Manual	7	5.34%
No control	25	19.08%

**Table 17 diagnostics-11-01731-t017:** The PUUS values of patients regarding the type of uterine cavity control.

Type of Uterine Control	Mean Values	95% CI
Instrumental	0.45	0.29–0.62
Manual	0.14	0.21–0.49
No control	0.80	0.34–1.26

**Table 18 diagnostics-11-01731-t018:** The proportions of the PUUS values of patients regarding the type of uterine cavity control.

Type of Uterine Control	0	1	2	3	4
Instrumental	73.74%	11.11%	11.11%	4.04%	0.00%
Manual	85.71%	14.29%	0.00%	0.00%	0.00%
No control	52.00%	32.00%	4.00%	8.00%	4.00%

**Table 19 diagnostics-11-01731-t019:** The proportions of the PUUS values of patients regarding the gender of the neonate.

Gender	0	1	2	3	4	Total
Male	44	11	7	3	1	66
	66.66%	16.66%	10.60%	4.54%	1.51%	100.00%
Female	48	9	5	3	0	65
	73.84%	13.84%	7.69%	4.61%	0.00%	100.00%
Total	92	20	12	6	1	131
	70.22%	15.26%	9.16%	4.58%	0.76%	100.00%

**Table 20 diagnostics-11-01731-t020:** The proportions of PUUS values of patients regarding the presence/absence of placenta previa, after removing the only case of PUUS = 4 and the few cases of PUUS = 3 (6 cases).

Placenta Previa	0	1	2	Total
No	83	17	12	112
	74.10%	15.17%	10.71%	100.00%
Yes	9	3	0	12
	75.00%	25.00%	0.00%	100.00%
Total	92	20	12	124
	74.19%	16.12%	9.67%	100.00%

**Table 21 diagnostics-11-01731-t021:** The proportions of the PUUS values of patients regarding the postpartum bleeding. There was no patient with heavy bleeding. Most patients had a low level of bleeding, and only a few had moderate bleeding.

Bleeding	0	1	2	3	4	Total
Low	83	16	11	5	1	116
	71.55%	13.79%	9.48%	4.31%	0.86%	100.00%
Moderate	9	4	1	1	0	15
	60.00%	26.66%	6.66%	6.66%	0.00%	100.00%
Total	92	20	12	6	1	131
	70.22%	15.26%	9.16%	4.58%	0.76%	100.00%

**Table 22 diagnostics-11-01731-t022:** The proportion of the patients’ PUUS values regarding the gestational age of the parturients. There were three patients whose gestational age was not known because they arrived during the fetal expulsion period, and they had not been surveyed during pregnancy.

Gestational Age (Weeks)	0	1	2	3	4	Total
33	0	0	1	0	0	1
	0%	0%	100%	0%	0%	100%
34	1	1	0	0	0	2
	50%	50%	0%	0%	0%	100%
35	2	0	0	0	0	2
	100%	0%	0%	0%	0%	100%
36	5	1	0	0	0	6
	83.33%	16.66%	0%	0%	0%	100%
37	9	5	2	1	0	17
	52.94%	29.41%	11.76%	5.88%	0%	100%
38	29	5	2	3	0	39
	74.35%	12.82%	5.12%	7.69%	0%	100%
39	35	6	6	2	1	50
	70%	12%	12%	4%	2%	100%
40	10	0	0	0	0	10
	100%	0%	0%	0%	0%	100%
41	1	0	0	0	0	1
	100%	0%	0%	0%	0%	100%
NK	1	2	0	0	0	3
	33.33%	66.66%	0%	0%	0%	100%
Total	93	20	11	6	1	131
	70.99%	15.26%	8.39%	4.58%	0.76%	100%

NK, gestational age not known, because the parturients arrived at the hospital during the fetal expulsion period, and they had not been surveyed during pregnancy.

## Data Availability

Datasets of patients, without names, are available from the corresponding author upon reasonable request.

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
