# Peer review of "The Importance of the Novel Postpartum Uterine Ultrasonographic Scale in Numerical Assessments of Uterine Involution Regarding Perinatal Maternal and Fetal Outcomes"

_diagnostics, 2021, doi:10.3390/diagnostics11091731_

Round 1
Reviewer 1 Report
Although the article has improved substantially, there are still many grammatical errors. Some ideas are difficult to understand, and the discussion and conclusion are not well developed.
In addition, as indicated in the first round of revision, this is not a second part.
More importantly, the article: ‘’Covali R, Socolov D, Socolov RV, Akad M. Postpartum uterine ultrasonographic scale: a novel method to standardize the assessment of uterine postpartum involution. J Med Life (in press) ‘’ is not citable, because it does not have an DOI.
Comments by sections
Material and method
Line 121: The PUUS method, described previously [16??],
Are figures 1 to 5 coming from the previous article? I think this article is not citable because it does not have a DOI.
Covali R, Socolov D, Socolov RV, Akad M. Postpartum uterine ultrasonographic scale: a novel method to standardize the assessment of uterine postpartum involution. J Med Life (in press)
Lines 152-154: the goal or aim of the study should be in the Introduction section (not in the MyM)
Results
Line 216: could you include the p-value in table 3, please
Line 217: I did not find in the manuscript the description of table 4.
Line 238: I suggest avoiding speculations (seemed?). It was or it was not different according to statistics
Line 363: in our study? I would suggest this study.
Lines 379-386: This paragraph is difficult to understand
Line 378-408: Don't understand the reason to discuss active management
Conclusion section
It must be changed because it should not be a list of the results previously stated
Author Response
Response to REVIEWER 1
Open Review
(x) I would not like to sign my review report
( ) I would like to sign my review report
English language and style
(x) Extensive editing of English language and style required
( ) Moderate English changes required
( ) English language and style are fine/minor spell check required
( ) I don't feel qualified to judge about the English language and style
|
Yes |
Can be improved |
Must be improved |
Not applicable |
|
|
Does the introduction provide sufficient background and include all relevant references? |
( ) |
(x) |
( ) |
( ) |
|
Is the research design appropriate? |
( ) |
( ) |
(x) |
( ) |
|
Are the methods adequately described? |
( ) |
( ) |
(x) |
( ) |
|
Are the results clearly presented? |
( ) |
( ) |
(x) |
( ) |
|
Are the conclusions supported by the results? |
( ) |
( ) |
(x) |
( ) |
Comments and Suggestions for Authors
Although the article has improved substantially, there are still many grammatical errors. Some ideas are difficult to understand, and the discussion and conclusion are not well developed.
-We corrected them. We have the certificate.
In addition, as indicated in the first round of revision, this is not a second part.
-We REMOVED was was highlighted in yellow, in line 114:
„The PUUS method, described in extenso in the first related article [16], was a visual scale to evaluate the number”
and in line 125 we REMOVED what was highlighted in yellow:
„The PUUS method, described in extenso in the first article [16], evaluated the proportion of the endometrial length”
and in Conclusion, we REMOVED what was highlighted in yellow: ”in this second part of our study we precisely demonstrated...”
More importantly, the article: ‘’Covali R, Socolov D, Socolov RV, Akad M. Postpartum uterine ultrasonographic scale: a novel method to standardize the assessment of uterine postpartum involution. J Med Life (in press) ‘’ is not citable, because it does not have an DOI.
-We REMOVED this article from References, and now number 16 in References is the following article:
- “Mulic-Lutvica A, Bekuretsion M, Bakos O, Axelsson O. Ultrasonic evaluation of the uterus and uterine cavity after normal, vaginal delivery. Ultrasound Obstet Gynecol. 2001 Nov;18(5):491-8. doi: 10.1046/j.0960-7692.2001.00561.x.”
Comments by sections
Material and method
Line 121: The PUUS method, described previously [16??],
-We corrected, [16] instead of [8] in line 121, as follows:
„The PUUS method, described in extenso in the first article [16],”,
which corresponds, indeed, to reference number 16:
- Covali R, Socolov D, Socolov RV, Akad M. Postpartum uterine ultrasonographic scale: a novel method to standardize the assessment of uterine postpartum involution. J Med Life (in press)
and then we REMOVED this article , as you suggested above.
Are figures 1 to 5 coming from the previous article? I think this article is not citable because it does not have a DOI.
Covali R, Socolov D, Socolov RV, Akad M. Postpartum uterine ultrasonographic scale: a novel method to standardize the assessment of uterine postpartum involution. J Med Life (in press)
-No, figures 1 to 5 can be found only in this article. The previous article included drawings. And we removed the previous article from References.
Lines 152-154: the goal or aim of the study should be in the Introduction section (not in the MyM)
-We did move them upward, at the end of introduction, lines 109-111:
„The goal of this related part of our work was to use this novel PUUS scale in order to assess the uterine cavity involution depending on maternal and fetal outcomes, on the way of delivery and medication received.”
Results
Line 216: could you include the p-value in table 3, please
-We did include the p-value in the table 3, and highlighted it in the text above table 3, too, as follows:
The PUUS grade varied significantly (P = .002) with the method of delivery of the baby: vaginal delivery (0.84±1.11, 95%CI: 0.5-1.18) or cesarean delivery (0.33±0.71, 95%CI: 0.18-0.48). Vaginal delivery was associated with a significantly slower involution of the uterine cavity compared with the cesarean delivery (Table 3).
Table 3. The PUUS values of patients regarding the method of delivery (P = .002).
|
Method of delivery |
Mean values |
95%CI |
|
Cesarean |
0.33 |
0.18-0.48 |
|
Vaginal |
0.84 |
0.50-1.18 |
We also removed from line 215 the word „(Table 2)” and from line 216 the word „(Figure 1)”, to make the text clear.
Line 217: I did not find in the manuscript the description of table 4.
-We added lines 223-226, as highlighted in yellow:
„The number and percent of patients, who delivered either by cesarean section or vaginally, for every value of PUUS, are detailed in Table 4. The most numerous patients (69; 79.31%) delivered by cesarean section and had a PUUS grade of 0, the least numerous patients (1; 2.27%) delivered vaginally and had a PUUS grade of 4.”
Line 238: I suggest avoiding speculations (seemed?). It was or it was not different according to statistics
-We rephrased it, as follows:
„In patients of rural origin (n=88, 67.17%), uterine cavity involution was not significantly slower (P = .24) than in patients of urban origin (n=43, 32.83%).”
Line 363: in our study? I would suggest this study.
-We did correct „this study”, as follows:
„In this study, there were 27 women...”
Lines 379-386: This paragraph is difficult to understand
-We removed it, as you recommended below.
Line 378-408: Don't understand the reason to discuss active management
We removed former lines 378-408 about active management, and the corresponding former titles 26-30 from References.
Conclusion section
It must be changed because it should not be a list of the results previously stated
-We changed it, as highlighted in yellow:
By standardizing the uterine involution in a numerical fashion (PUUS), in this study we precisely demonstrated that the uterine involution was slower in vaginal deliveries, and increased with the number of vials of oxytocin received intrapartum. No significant influence was associated with any of the other conditions studied. This PUUS scale could farther be used to study the influence of a variety of factors upon the speed of uterine involution and may improve the quality of postpartum care.
Submission Date
16 August 2021
Date of this review
04 Sep 2021 04:54:38

Reviewer 2 Report
The manuscript entitled “The importance of the novel postpartum uterine ultrasonographic scale in assessment in a numerical fashion of the uterine involution as regards perinatal maternal and fetal outcomes” by Roxana Covali and colleagues describes the use of the PUUS (Postpartum Uterine Ultrasonographic Scale) method evaluating the length of the endometrium of the uterine cavity occupied by blood or debris after delivery.
The comments to this manuscript have been reported below.
- A summary table of the clinical characteristics of the patients is not presented so it is not clear which kind of patients the authors studied. For example, are there patients with coagulopathy? uterine inversion? Moreover, obesity is also a recognized risk factor for postpartum uterine atony playing an important role in uterine involution but Body mass index (BMI) of the patients is not reported.
- Since the authors analysed only 1 patients with PUUS 4, PUUS 4 should be removed from the study because no statistical analysis can be made in this group
- Due to the observatory-related variability that can occur in this type of study, evaluations of ultrasonography measurements should be performed independently by at least two observers and the level of concordance, expressed as the percentage of agreement between the observers, should be reported.
Author Response
Response to REVIEWER 2
Open Review
( ) I would not like to sign my review report
(x) I would like to sign my review report
English language and style
( ) Extensive editing of English language and style required
(x) Moderate English changes required
-We edited it. We have the certificate.
( ) English language and style are fine/minor spell check required
( ) I don't feel qualified to judge about the English language and style
|
Yes |
Can be improved |
Must be improved |
Not applicable |
|
|
Does the introduction provide sufficient background and include all relevant references? |
( ) |
( ) |
(x) |
( ) |
|
Is the research design appropriate? |
( ) |
(x) |
( ) |
( ) |
|
Are the methods adequately described? |
( ) |
(x) |
( ) |
( ) |
|
Are the results clearly presented? |
( ) |
( ) |
(x) |
( ) |
|
Are the conclusions supported by the results? |
( ) |
( ) |
(x) |
( ) |
Comments and Suggestions for Authors
The manuscript entitled “The importance of the novel postpartum uterine ultrasonographic scale in assessment in a numerical fashion of the uterine involution as regards perinatal maternal and fetal outcomes” by Roxana Covali and colleagues describes the use of the PUUS (Postpartum Uterine Ultrasonographic Scale) method evaluating the length of the endometrium of the uterine cavity occupied by blood or debris after delivery.
The comments to this manuscript have been reported below.
- A summary table of the clinical characteristics of the patients is not presented so it is not clear which kind of patients the authors studied. For example, are there patients with coagulopathy? uterine inversion? Moreover, obesity is also a recognized risk factor for postpartum uterine atony playing an important role in uterine involution but Body mass index (BMI) of the patients is not reported.
-We added, at the end of Discussion, lines 420-422, as highlighted in red:
”Larger studies to show the correlation between uterine involution and other specific factors (clinical characteristics of the patients, coagulopathies, uterine inversions, Body mass index), with more numerous patients, are also required.”
- Since the authors analysed only 1 patients with PUUS 4, PUUS 4 should be removed from the study because no statistical analysis can be made in this group
-We added, as highlighted in red, after Table 1:
„Table 1. Distribution of patients regarding the PUUS value.
|
PUUS value |
Number of patients |
Percent |
|
0 |
92 |
70.22% |
|
1 |
20 |
15.26% |
|
2 |
12 |
9.16% |
|
3 |
6 |
4.58% |
|
4 |
1 |
0.76% |
|
Total |
131 |
100% |
PUUS grade varied with age, but there was no statistically significant correlation between PUUS grade and age of the parturients (P= .51) (Figure 6).
Since there was only 1 patient with PUUS 4, and no statistical analysis could be made with this group, he was removed from the statistical analysis, and remained only some tables of figures.”
- Due to the observatory-related variability that can occur in this type of study, evaluations of ultrasonography measurements should be performed independently by at least two observers and the level of concordance, expressed as the percentage of agreement between the observers, should be reported.
-We added, as highlighted in red:
“To eliminate any potential sources of bias: 1) patients were examined in a random order, and the practitioner was unaware of the patient’s medical history at the time of the examination, and 2) there was only one practitioner (RC) who performed all the ultrasonography measurements, under the supervision of RS, evaluation was performed independently by both of them, level of concordance 95.41%, and the disagreements were solved by discussion.”
Submission Date
16 August 2021
Date of this review
29 Aug 2021 21:58:58

Reviewer 3 Report
I recommend minor changes to the way the subject is presented. Also, the avoidance of self-citations is necessary (example: bibliographic item 23)...
Author Response
Response to REVIEWER 3
Open Review
(x) I would not like to sign my review report
( ) I would like to sign my review report
English language and style
( ) Extensive editing of English language and style required
( ) Moderate English changes required
( ) English language and style are fine/minor spell check required
(x) I don't feel qualified to judge about the English language and style
|
Yes |
Can be improved |
Must be improved |
Not applicable |
|
|
Does the introduction provide sufficient background and include all relevant references? |
( ) |
(x) |
( ) |
( ) |
|
Is the research design appropriate? |
(x) |
( ) |
( ) |
( ) |
|
Are the methods adequately described? |
(x) |
( ) |
( ) |
( ) |
|
Are the results clearly presented? |
(x) |
( ) |
( ) |
( ) |
|
Are the conclusions supported by the results? |
(x) |
( ) |
( ) |
( ) |
Comments and Suggestions for Authors
I recommend minor changes to the way the subject is presented.
-We did correct them. We have the certificate.
Also, the avoidance of self-citations is necessary (example: bibliographic item 23)...
-We REMOVED the self-citation, bibliographic number 23, former lines 348-349: „ or even a combined endoscopic-ultrasound approach [23]”, and from the References we removed:
- Socolov R, Tanos V, Butureanu T, Akad M, Bumbu G, Socolov D. Rare cervico-vaginal malformation treated by hystero-laparoscopy and urological approach. Proceedings of the 14th National Congress of Urogynecology and The National Conference of The Romanian Association for the Study of Pain. Eforie, Romania. 2017:241-245 (ISBN:978-88-95922-98-0).
-We also REMOVED the other article that could be considered self-citation, which was number 16:
- Covali R, Socolov D, Socolov RV, Akad M. Postpartum uterine ultrasonographic scale: a novel method to standardize the assessment of uterine postpartum involution. J Med Life (in press)
Submission Date
16 August 2021
Date of this review
23 Aug 2021 20:42:10

Round 2
Reviewer 1 Report
The study has improved significantly. Congratulations!
Reviewer 2 Report
Although the manuscript has been improved, there are still some points that can be improved such as:
- at least the clinical characteristics of the patients studied should be reported
- PUUS 4 grade should be removed from the manuscript as it can NEVER be statistically analyzed (being a single case). This makes difficult to understand the data reported in the tables (too many) making the manuscript confusing and difficult to interpret.
However, if the authors do not have these data and can not improve the data exposition, the manuscript can be accepted in the present form.
This manuscript is a resubmission of an earlier submission. The following is a list of the peer review reports and author responses from that submission.
Round 1
Reviewer 1 Report
The objective of this study was to analyze the uterine involution between patients who underwent cesarian and vaginal delivery, as also the impact of socioeconomic status, the effects of oxytocin or ergometrine treatments, and postpartum uterine curettage on uterine involution features
Globally, the article is relevant in the area and it is reporting the importance of different factors associated with delivery methods and postpartum treatments that can influence uterine involution. The study design and methodology are appropriated. However, the article title needs to be focused on the main findings, as also it is lacking a clear conclusion.
Comments by sections
Material and method
- Could you indicate how it was defined the rural and urban residencies? i.e. starting 5 km from…
- Could be the socioeconomic status more important than the type of residency?
Discussion
- Discussion section must be improved. Only mentioning one study per paragraph is not the right format for a discussion.
You can check a discussion example in the following article: Imai K, Kotani T, Tsuda H, Nakano T, Hirakawa A, Kikkawa F. A Novel Approach to Detecting Postpartum Hemorrhage Using Contrast-Enhanced Ultrasound. Ultrasound Med Biol. 2017 Mar;43(3):615-620. doi: 10.1016/j.ultrasmedbio.2016.11.008. Epub 2016 Dec 23. PMID: 28024660.
Minor comments
-Could you include more information about PUUS methodology in the introduction section please? i.e. the meaning of the acronym
-I would include line 82-83 in the introduction
-Line 197: I suggest indicating that: Bardin et al (9)
-this is not the second part of your study…… you can mention your first study, but this one is not a second part, it is a related one but independent from the previous one. In other words, for this study the first part would be the analysis of method delivery and you second one the effects of pharmacological treatments.
Reviewer 2 Report
This study evaluates the postpartum uterine involution using ultrasound scale. It is an interesting idea however this manuscript is problematic for many reasons. From the methods section it is not clear how the measurements were performed and how the patient is managed during the third stage of labour. It is also not clear whether prophylactic uterotonics are offered to all patients in the third stage of labour. When during the labour did the patients receive oxytocin and ergometrine? How would the authors justify that 75% of all participants had curettage performed after delivery?? In addition lines 89-94 should be in the Results section. The discussion section is poorly written. What would be most interesting and most clinically applicable would be to evaluate how the uterine involution correlates with the amount of lost blood?
Round 2
Reviewer 2 Report
This is a revised manuscript about the evaluation of uterine involution post-partum using an ultrasound scale. The authors still do not provide the justification of their management of the third stage of labour. The methods are poorly written. The authors state that the hospital policy requires expectant management of the third stage of labour. Then in the discussion they cite RCOG guidelines which provide detailed protocols about the management of minor and major PPH. The authors still do not provide an explanation about the unacceptably high rate of curettage after delivery. The discussion should be more balanced. Just citing certain research is not enough. A critical assessment of available literature is recommended as well as comparison of authors’ hospital protocol to international guidelines and recommendations.